# VSP: Assessing the dual challenges of perception and reasoning in spatial planning tasks for MLLMs

## Abstract

Multimodal large language models are an exciting emerging class of language models (LMs) that have merged classic LM capabilities with those of image processing systems. However, the ways that these capabilities combine are not always intuitive and warrant direct investigation. One understudied capability in MLLMs is *visual spatial planning*—the ability to comprehend the spatial arrangements of objects and devise action plans to achieve desired outcomes in visual scenes. In our study, we introduce **VSP**, a benchmark that 1) evaluates the spatial planning capability in these models in general, and 2) breaks down the visual planning task into finer-grained sub-tasks, including perception and reasoning, and measure the capabilities of models in these sub-tasks. Our evaluation shows that both open-source and private MLLMs fail to generate effective plans for even simple spatial planning tasks. Evaluations on the fine-grained analytical tasks further reveal fundamental deficiencies in the models' visual perception and bottlenecks in reasoning abilities, explaining their worse performance in the general spatial planning tasks. Our work illuminates future directions for improving MLLMs' abilities in spatial planning.

## 1 Introduction

The rapid advancement of large language models has driven considerable growth in their capabilities to produce fluent text in many domains, generating outputs exhibiting potential "reasoning" and "understanding" abilities. Touvron et al. (2023); Bi et al. (2024); Jiang et al. (2024); Brown et al. (2020). Recently, multimodal large language models (MLLMs) have advanced on LLMs through additional training on native image inputs, to achieve impressive performance generating text describing and relating to input images Achiam et al. (2023); Liu et al. (2024); Team et al. (2023); Awadalla et al. (2023); Alayrac et al. (2022), with applications in image captioning, visual question answering, visual reasoning, and others Ying et al. (2024); Yang et al. (2024); Shao et al. (2023); Zheng et al. (2023). The swift evolution of MLLMs has enabled them to tackle increasingly sophisticated tasks that require multiple emerging abilities in complex scenarios. However, as model capabilities and deployment needs advance, the challenges in usefully evaluating them grow in kind.

Planning is a fundamental capability in intelligent systems that is particularly contested in LMs Valmeekam et al. (2023), and is understudied in MLLMs. Visual spatial planning refers to the task of comprehending the spatial arrangement of objects in a scene and designing action plans to achieve a desired outcome. For example, the classical maze problem can be considered a visual planning task, where an agent is given an input image describing the maze environment and is asked to produce a viable path to navigate the player from the starting position to the goal. This task requires two capabilities: *image perception*, which enables the agent to understand the objects, environment and spatial relations present in the image, and *reasoning*, which enables the agent to perform strategic decision-making.

Visual spatial planning is an important capability in many potential applications for MLLMs, such as navigating in complex environments with autonomous driving Tian et al. (2024); Ma et al. (2023) or manipulating objects with robotic hands Chang et al. (2023); Hu et al. (2023). However, although there have been increasingly more benchmarks to evaluate the vision processing capabilities of MLLMs, few current benchmarks systematically evaluate their capability of performing visual spatial

Table 1: Comparison with representative existing benchmarks.

| Name | Tasks Description | Keywords |
|---|---|---|
| **MME** Fu et al. (2023) | Image content understanding, reasoning | perception, reasoning |
| **MMMU** Yue et al. (2024) | College-level knowledge reasoning | multi-discipline knowledge, reasoning |
| **MathVision** Wang et al. (2024) | Math problems with visual contexts | mathematical reasoning |
| **SeedBench** Li et al. (2023a) | Comprehension of scene & instance in image | perception, reasoning, spatial relation |
| **MM-Vet** Yu et al. (2023) | General problems that need integrated abilities | perception, reasoning, spatial relation |
| **VSP** | Understand & extract *spatial* info and *plan accordingly* | *Spatial* planning, *Spatial* perception, reasoning |

planning tasks. As shown in Table 1, existing benchmarks mostly focus on MLLMs' ability to understand image content and perform visual logic reasoning Fu et al. (2023); Yue et al. (2024); Wang et al. (2024); however, they often overlook the ability to comprehend the spatial arrangements of entities within images and to devise spatial action plans based on practical restrictions in the visual environment. As a result, two research questions are left unanswered: ❶ How performant are MLLMs in performing visual planning tasks? ❷ What are the bottleneck capabilities, *e.g.*, perception or reasoning, that limit the performance of MLLMs in visual planning tasks?

To this end, we introduce Visual Spatial Planning (VSP), a benchmark specifically designed to evaluate the spatial planning capabilities of MLLMs. As illustrated in Figure 1, the VSP benchmark consists of various environment settings designed to assess the capabilities of models in different scenarios. The first two scenarios, *Maze Navigation* and *Blocks World*, are basic environments developed from classical planning tasks. Additionally, we challenge the model's abilities in dynamic and realistic applications through the *Collision* and *Google Map* scenarios, respectively. All environments are fully observable through input images. The MLLMs are required to

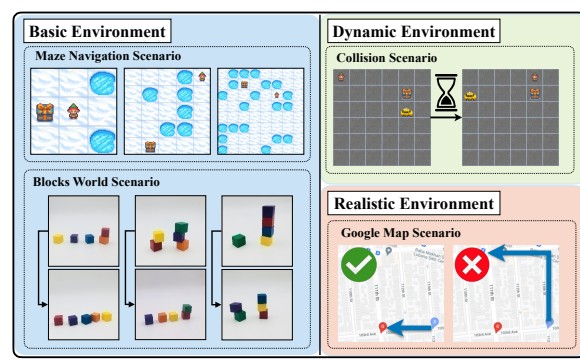

Figure 1: The overview of the proposed visual spatial planning benchmark.

interpret the visual inputs, deduce the consequences of each action, and execute the designated tasks. To comprehensively evaluate the fine-grained capabilities needed for visual spatial planning, VSP includes 4.6K questions in 10 meticulously designed tasks that feature both simulated and photo-realistic visual inputs. In addition to testing end-to-end spatial planning, these tasks test essential individual visual planning abilities, such as image perception and reasoning.

We apply the VSP benchmark to evaluate existing state-of-the-art MLLMs, including both open-source and private ones. Surprisingly, we find that even the most competitive MLLMs sometimes struggle in performing the simplest visual planning tasks, such as a 3x3 maze problem. Our fine-grained capability analysis further reveals that existing MLLMs have flaws in reasoning and bigger bottlenecks in perception. We believe the VSP benchmark highlights critical weaknesses in current MLLMs and sheds light on future directions for enhancing their spatial understanding and planning capabilities.

## 2 RELATED WORK

### 2.1 GENERAL PLANNING IN LMS

Planning has been a central focus of research in AI. Traditional work in AI planning includes using formal languages to represent and solve planning problems Aeronautiques et al. (1998), and developing algorithms like dynamic programming and reinforcement learning to explore environments and formulate viable plans Guo et al. (2014); Sutton (1991). While these works mostly focus on predefined and restricted environments, recently, with the advancement of LMs, it has become intriguing to study whether LMs, with the potential to be general intelligent agents, can perform

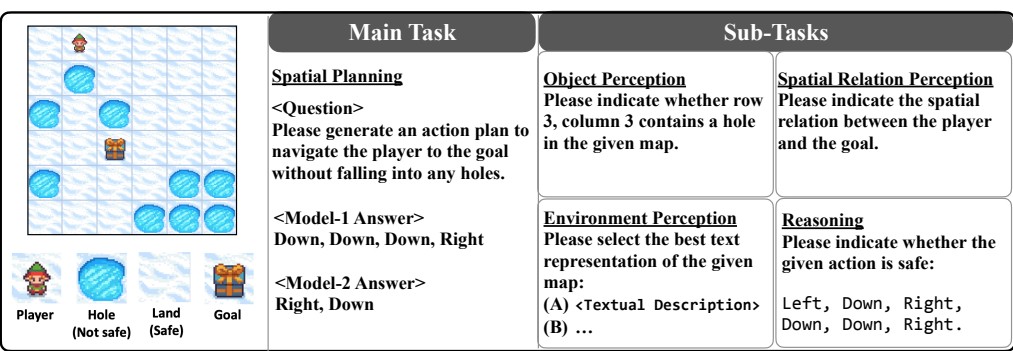

Figure 2: Overview of the *Maze Navigation* scenario.

planning in different settings and environments Kambhampati (2024); Kambhampati et al. (2024); Stechly et al. (2024). Many works explore the best ways to activate the planning capabilities of LMs, including divide and conquer Wei et al. (2022); Yao et al. (2024); Shen et al. (2024); Yao et al. (2022), grounding outputs in admissible actions Ahn et al. (2022); Hazra et al. (2024), retrospecting and refining Shinn et al. (2024); Madaan et al. (2024), and leveraging external tools Guan et al. (2023); Ruan et al. (2023). Meanwhile, with the increasing capabilities of LMs, growing research efforts are now dedicated to benchmark their planning capabilities in various complex environments Wu et al. (2023); Xie et al. (2024); Valmeekam et al. (2022).

## 2.2 SPATIAL AND VISUAL PLANNING IN LMs

Many general planning tasks in LMs involve understanding visual environments and comprehending spatial information. In robotics and embodied agent studies, LMs play a crucial role in grounding visual entities with references in open-domain instructions and formulating plans based on spatial constraints. Consequently, they are increasingly used in physically grounded scenarios such as object rearrangement Chang et al. (2023); Hu et al. (2023), cooking Joublin et al. (2023); Sakib and Sun (2023), and navigation Hazra et al. (2024); Ahn et al. (2022). LMs are also used in AIGC to propose spatial arrangements of entities following instructions Feng et al. (2024). While realistic planning tasks align with real needs, their complexity and expansive action spaces limit the analysis of LMs' detailed planning capabilities. Therefore, research also focuses on LMs' planning in simulated environments and games. For example, *mystery blocksworld* is a dynamically generated set of blocksworld tasks to test generalization in LMs Valmeekam et al. (2023). Additionally, many text games have been introduced to test LMs' abilities in spatial understanding and imagination Shridhar et al. (2020); Wu et al. (2023); Yang et al. (2023); Aghzal et al. (2023). However, most of these studies transform visual information into text inputs, thus not directly measuring LMs' visual abilities.

## 2.3 BENCHMARKS FOR MLLMs

MLLMs have inherited and advanced many intriguing features from text-only LMs Yang et al. (2023); Qi et al. (2023). Benchmarks for MLLMs have rapidly emerged to evaluate performance in areas such as image content understanding Fu et al. (2023); Cha et al. (2024), perception Ge et al. (2023); Tong et al. (2024), knowledge Yue et al. (2024); Wang et al. (2024); Lu et al. (2023), and reasoning Fu et al. (2023); Yue et al. (2024); Liu et al. (2023). While these benchmarks quantify MLLMs' abilities in many fields, their capabilities in spatial understanding and reaction are relatively under-explored. Some benchmarks cover spatial relations understanding Li et al. (2023a); Yu et al. (2023), but often overlook the ability to devise complex spatial action plans based on visual environment constraints. We focus on *visual spatial planning* - the ability to comprehend spatial arrangements of objects and devise action plans to achieve specific outcomes. We fill the gap in benchmarking MLLM abilities for visual spatial planning and highlight future directions for improving MLLMs towards models with general intelligence.

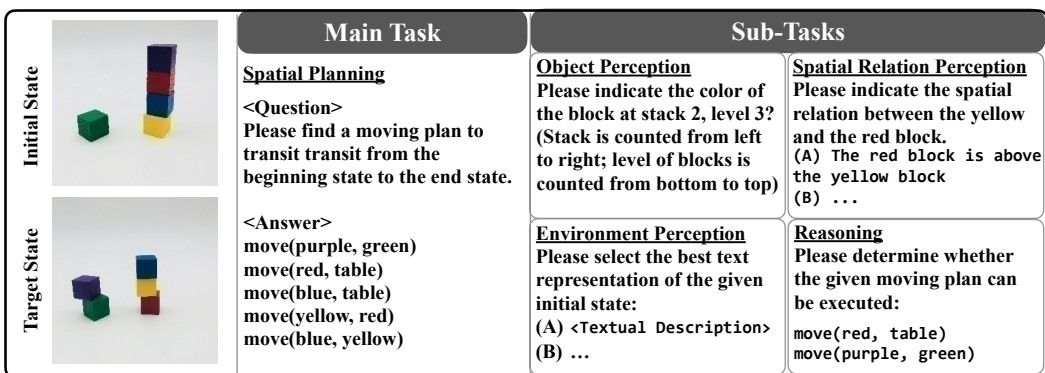

Figure 3: Overview of the *Blocks World* scenario.

# 3 THE VISUAL SPATIAL PLANNING BENCHMARK

## 3.1 OVERVIEW OF THE BENCHMARK

In this benchmark, our objectives are two-fold: ❶ quantify the visual spatial planning capabilities of current MLLMs; and ❷ uncover current capability bottlenecks that limit the effectiveness of MLLMs in visual spatial planning tasks. The first objective requires *broader* task designs. Specifically, the tasks should range from classical planning tasks Brockman et al. (2016); Valmeekam et al. (2022) to ones in those more dynamic and realistic environments. On top of that, the second objective requires *deeper* task designs. In particular, performing spatial planning in visual environments requires a series of cohesive steps. For example, to generate an accurate path to navigate a player to a goal, an agent needs to be able to correctly view and understand the visual map, reason to find which actions are safe or dangerous, and come up with a detailed plan to achieve the goal. Each of these steps could be challenging for a developing MLLM, and understanding which of these subtasks challenge them most will drive future improvement.

To this end, we propose the Visual Spatial Planning (VSP) benchmark, with the objective of measuring and diagnosing the capabilities of MLLMs in producing accurate spatial plans in visual environments. The VSP benchmark consists of four scenarios: ❶ the simulated **Maze Navigation scenario**, whose main task is to move a game character through a maze; ❷ the photo-realistic **Blocks World scenario**, whose main task is to move blocks from a starting configuration to a goal configuration; ❸ the dynamic **Collision scenario**, whose main task is to determine if there is a danger of collision between two objects in the environment, and ❹ the realistic **Google Map scenario**, whose main task is to find a path in real streets of New York City. In addition to the main task, VSP introduces four sub-tasks in first two scenarios that focus on the individual capabilities needed for the main task:

- **T1. Single Object Perception** – Determine the characteristics of a single object;
- **T2. Spatial Relation Perception** – Determine the relative positions of two objects;
- **T3. Environment Perception** – Find textual descriptions that describe the visual environment;
- **T4. Spatial Reasoning** – Determine the consequence of a series of actions or moves.

The sub-task details are designed specific to both scenarios. Furthermore, to demonstrate the model's performance under different levels of environmental complexity, we establish progressive difficulty settings for various tasks, which are measured by parameters such as map size, minimum required number of actions, *etc.* The detailed task statistics are provided in appendix A. In what follows, we introduce each scenario in detail, as well as the data curation and the task creation processes.

## 3.2 THE MAZE NAVIGATION SCENARIO

The *Maze Navigation* scenario is inspired by the popular implementation Brockman et al. (2016) of a fully observable path-finding problem. As depicted in Figure 2 left, it simulates a classical grid world environment with a designated start and goal position, where part of the grids contain obstacles (the "holes") and cannot be passed through.

Figure 4: Benchmark creation process. **Left:** We prepare input images that fulfill the task requirements with different difficulties. **Mid:** We formulate input prompts for each task. The input prompts consists of interleaved texts and images. **Right:** We develop automatic evaluation process for each task.

The main spatial planning task and the four sub-tasks are defined as follows:

- **Main Task** (Spatial Planning) – Generate a safe path to navigate from the start grid to the goal;
- **T1** (Single Object Perception) – Determine if a specified grid is safe;
- **T2** (Spatial Relation Perception) – Find spatial relations between the player and the goal;
- **T3** (Environment Perception) – Find the textual description that fits the visual environment;
- **T4** (Spatial Reasoning) – Determine the consequence of a given action series.

An example of input image and questions is demonstrated in Figure 2. Each task is equipped with progressive adjusted difficulty settings to evaluate the model's capability under various circumstances. For **Main Task** and **T1**-**T3**, the difficulties are measured by the size of the map, ranging from 3x3 to 8x8, where a larger map introduces more challenges in correctly perceiving objects and planning accordingly. For task **T4**, since a longer path naturally introduces more challenges for reasoning, we adopt path length ranging from 1 to 9 as the difficulty measure. Please refer to Appendix A for the complete example of the question and answer in each task.

## 3.3 THE BLOCKS WORLD SCENARIO

The *Blocks World* is a widely-adopted planning problem Valmeekam et al. (2022); Hao et al. (2023); Gokhale et al. (2019). As depicted in Figure 3 left, in this scenario, the agent is given images containing sets of blocks in unique colors. These blocks are stacked vertically, forming multiple stacks on the table. The agent is asked to turn the blocks from initial state to target state through a series of moving actions. For each action, the agent can only move the top block of any stack, providing it is moved to either the table or the top of another stack.

Similarly, the main spatial planning task and the four sub-tasks are defined as follows:

- **Main Task** (Spatial Planning) – Form a moving plan to achieve the target state of block arrangement;
- **T1** (Single Object Perception) – Determine the color of the block at a specific position;
- **T2** (Spatial Relation Perception) – Determine the spatial relation between two blocks;
- **T3** (Environment Perception) – Find the text representation that fits the visual environment;
- **T4** (Spatial Reasoning) – Determine the consequence of a given moving plan.

An example of input image and questions is demonstrated in Figure 3. Similar to the *Maze Navigation* scenario, each task is equipped with progressive adjusted difficulty. Specifically, in **Main Task** and **T4**, the difficulties are measured by the number of actions involved, ranging from 1 to 7, which quantifies the complexity of the action plan. On the other hand, for tasks **T1**-**T3**, which focus on

perception, the difficulty is measured by the number of blocks presented in the image, ranging from 3 to 5. Please refer to Appendix A for the complete example of the question and answer in each task.

### 3.4 THE COLLISION SCENARIO AND GOOGLE MAP SCENARIO

On top of the previous two scenarios, we further experiment in a more dynamic (the collision scenario) and realistic (the Google map scenario) settings to explore the capabilities of models in more challenging cases. The input of these two scenarios are shown in the right panels of Figure 1. Please refer to Appendix A for the complete example of each scenario.

• **Collision Scenario** In this scenario, the input map is similar to *Maze Navigation* scenario. However, in this case, the player is moving in an environment where a car is also present and moving. Given the moving information (speed, direction) of the player and the car, the model needs to determine the time the player and the car reaches the goal, and then determine if there is a collision danger here.

• **Google Map Scenario** In this scenario, the input image is a real Google map depicting the streets and avenues in New York City. The starting location and the goal are randomly chosen crossroads in the map and are marked on the image. The goal of the model is to find a path from the starting location to the goal. The model needs to output a path described by directions (north, east, west, south) and the number of blocks to traverse (e.g., head north for 2 blocks, then head east for 3 blocks).

### 3.5 BENCHMARK CREATION PROCESS

In Figure 4, we demonstrate the general process for benchmark creation. We use the *Blocks World* and *Maze Navigation* scenarios as examples.

First, in the left panel of Figure 4, we prepare the input images used for each task and scenario. In the *Maze Navigation* scenario, we generate input maps using the OpenAI Gym package Brockman et al. (2016), with modifications to ensure that the positions of the player, the goal, and the holes are all randomly generated. In the *Blocks World* scenario, we sample pairs of images from the BIRD dataset Gokhale et al. (2019), ensuring there is at least one viable plan to move the blocks from the initial state to the target state. The images are prepared conditional on different levels of difficulty.

Second, as shown in the center of Figure 4, we formulate input prompts for each task. The prompt consists of interleaved text and images to provide sufficient information. For example, for *Maze Navigation*, we include images to show the appearance of elements in the map and provide example maps to better illustrate how the models should interpret the map. We invite native speakers to refine the prompts so that they accurately describe the task requirements. The prompts are in Appendix A.

Finally, in the right panel of Figure 4, we evaluate the performance of MLLMs under each task. It is worth noting that the answer for each task is often not unique. For example, in the *Blocks World* scenario, there can be many ways to move the blocks to reach the target state. As such, we develop scripts to automatically evaluate the answers for each task.

In addition to the steps above, some tasks require extra steps to construct meaningful questions, candidates, and answers. For example, in T4 of the *Blocks World* scenario, the input actions must cover various valid/invalid movements. The detailed steps we followed to create each task set are provided in Appendix A. We release all images, texts, and scripts to facilitate replication and scaling.

## 4 EXPERIMENTS

In this section, we present evaluation results of state-of-the-art MLLMs under our main tasks and sub-tasks. Our goal is to answer the following research questions: ❶ How well can state-of-the-art MLLMs perform in the visual spatial planning tasks? ❷ What are the bottleneck capabilities that limit the MLLMs in visual spatial planning tasks?

### 4.1 BASELINES

We evaluate various representative MLLMs including both private and open-source models.

Table 2: Zero-shot success rates for the spatial planning task, at various difficulty levels. *Maze Navigation* difficulty levels represent the maze's square grid length. *Blocks World* difficulty levels correspond to the minimum number of steps to a solution. Results better than 30% are **bolded**.

| Difficulty level | MAZE NAVIGATION | | | | | | BLOCKSWORD | | | | COLLISION | GOO MAP | OVERALL |
|---|---|---|---|---|---|---|---|---|---|---|---|---|---|
| | 3 | 4 | 5 | 6 | 7 | 8 | 1 | 3 | 5 | 7 | | | |
| Gemini Team et al. (2023) | **0.31** | 0.26 | 0.15 | 0.06 | 0.14 | 0.10 | 0.10 | 0.14 | 0.00 | 0.01 | 0.13 | 0.00 | 0.1167 |
| GPT-Vision Achiam et al. (2023) | **0.55** | **0.36** | 0.27 | 0.13 | 0.17 | 0.10 | **0.50** | 0.17 | 0.03 | 0.00 | 0.24 | 0.02 | 0.2117 |
| Claude-3 AI (2024a) | **0.52** | **0.33** | 0.16 | 0.15 | 0.16 | 0.09 | 0.12 | 0.03 | 0.00 | 0.00 | 0.18 | 0.02 | 0.1467 |
| GPT-4o AI | **0.68** | **0.58** | **0.35** | 0.24 | 0.18 | 0.23 | **0.71** | **0.33** | 0.12 | 0.03 | 0.16 | 0.04 | **0.3042** |
| Pixtral AI (2024b) | **0.32** | 0.20 | 0.17 | 0.10 | 0.06 | 0.06 | 0.21 | 0.11 | 0.01 | 0.00 | 0.22 | 0.03 | 0.1242 |
| LLaVA Liu et al. (2024) | 0.03 | 0.03 | 0.02 | 0.08 | 0.09 | 0.04 | 0.04 | 0.01 | 0.00 | 0.00 | 0.02 | 0.00 | 0.0300 |
| InternLM Dong et al. (2024) | 0.27 | 0.16 | 0.06 | 0.05 | 0.04 | 0.07 | 0.10 | 0.03 | 0.00 | 0.00 | 0.25 | 0.00 | 0.0858 |
| InternLM-VL Dong et al. (2024) | 0.15 | 0.14 | 0.08 | 0.04 | 0.02 | 0.05 | 0.02 | 0.00 | 0.00 | 0.00 | 0.22 | 0.01 | 0.0600 |
| InstructBLIP Dai et al. (2024) | 0.00 | 0.00 | 0.00 | 0.00 | 0.00 | 0.00 | 0.00 | 0.00 | 0.00 | 0.00 | 0.00 | 0.00 | 0.0000 |
| SPHINX Lin et al. (2023) | 0.11 | 0.08 | 0.05 | 0.02 | 0.04 | 0.03 | 0.07 | 0.06 | 0.01 | 0.00 | 0.04 | 0.00 | 0.0425 |
| LLaMA-3.2 Meta (2024) | 0.23 | 0.18 | 0.16 | 0.08 | 0.08 | 0.10 | 0.05 | 0.00 | 0.00 | 0.00 | 0.14 | 0.00 | 0.0850 |

We cover the following *private models*: ① Gemini Team et al. (2023) has demonstrated remarkable capabilities in image understanding and reasoning. We adopt `Gemini-1.0-Pro-Vision` in our experiments ② GPT-4 Turbo with vision Achiam et al. (2023) inherent strong text understanding capabilities from GPT-4 and is equipped with vision capabilities. We use `turbo-2024-04-09` for evaluation. ③ Claude-3 AI (2024a) is a family of MLLMs strong at advanced reasoning and vision analysis. We adopt `claude-3-sonnet-20240229`, the default model used in chat interface and has comparable speed & cost with GPT Vision. ④ GPT-4o AI is a recently released MLLM with one of the most advanced abilities in processing combination of text, audio, and image outputs. We adopt `gpt-4o-2024-05-13` in experiments. ⑤ Pixtral AI (2024b) is a recent MLLM that demonstrates strong performance across a series of multimodal tasks. We adopt `pixtral-12b-2409` in our experiments. Additionally, we attempt to evaluate the latest OpenAI GPT o1 OpenAI (2024); however, the currently available version only supports text input, preventing us from evaluating it in our benchmark.

We cover the following *open-source models*: ⑥ LLaVA Liu et al. (2024) performs instruction tuning on LLaMA and projects image into text embedding space through CLIP Radford et al. (2021). We adopt LLAVA-V1.6-VICUNA-7B for evaluation. ⑦ InternLM-XComposer2 Dong et al. (2024) enhances ability to understand free-form text-image composition. The latest released checkpoints include `internlm-xcomposer2-7b` and `internlm-xcomposer2-vl-7b`, with the former focusing on general text-image composition and the latter focusing on VL benchmarks. We adopt both for evaluation. ⑧ InstructBLIP Dai et al. (2024) is a popular MLLM based on pre-trained BLIP-2 Li et al. (2023b) model. We adopt `blip2-t5-instruct-flant5xxl` for evaluation. ⑨ SPHINX Lin et al. (2023) unfreezes the LM during pre-training to enhance cross-model alignment. We adopt `SPHINX-v2-1k` for evaluation. ⑩ LLaMA-3.2 Meta (2024) is a recently released MLLM exceeding on image understanding tasks. We adopt `llama-3.2-90B-Vision` in our experiments. We use the public released checkpoints and codes.

## 4.2 MAIN TASK (SPATIAL PLANNING) EVALUATION

First, we present the main task evaluation results for the four scenarios, which reflect the general spatial planning capabilities of existing MLLMs. All the evaluation in this section is conducted under zero-shot setting without any fine-tuning or in-context learning. Evaluations with in-context learning and fine-tuning are presented in Sections 4.5 and 4.6.

The performance is demonstrated in Table 2. We also present difficulty levels in the table, which is measured by the size of the map (3 represents 3x3 maps) in *Maze Navigation* and by the minimum number of steps in *Blocks World*. From the table, we summarize our findings as follows:

**MLLMs have considerable room for improvement in spatial planning tasks.** We observe that both private and open-source models exhibit sub-optimal performance in various scenarios. In particular, open-source models face significant challenges and rarely succeed in these tasks. Besides, even the most capable private models could frequently make mistakes on relatively simple tasks, such as those involving a 3x3 size map or a single-step block moving task. Considering that these tasks would be

Table 3: Decomposed Capability Analysis. Similar to the spatial planning task, each task consists of test with different difficulties. Results better than 70% are **bolded**. Please refer to Appendix E for the complete evaluation results for different difficulties.

| Task | MAZE NAVIGATION | | | | BLOCKSWORD | | | |
|---|---|---|---|---|---|---|---|---|
| | **T1** | **T2** | **T3** | **T4** | **T1** | **T2** | **T3** | **T4** |
| Random Guess | 0.5 | 0.25 | 0.25 | 0.5 | 0.17 | 0.25 | 0.25 | 0.5 |
| Gemini Team et al. (2023) | 0.58 | 0.56 | 0.33 | 0.49 | **0.86** | 0.51 | 0.54 | 0.55 |
| GPT-Vision Achiam et al. (2023) | 0.56 | 0.27 | 0.46 | 0.56 | **0.73** | **0.80** | **0.70** | **0.71** |
| Claude-3 AI (2024a) | 0.45 | 0.67 | 0.32 | 0.61 | 0.43 | 0.53 | 0.49 | 0.66 |
| GPT-4o AI | 0.58 | 0.67 | 0.58 | **0.74** | **0.95** | **0.90** | **0.90** | **0.76** |
| Pixtral AI (2024b) | 0.44 | 0.35 | 0.33 | 0.51 | 0.49 | **0.72** | 0.63 | 0.57 |
| LLaVA Liu et al. (2024) | 0.49 | 0.27 | 0.21 | 0.54 | 0.22 | 0.21 | 0.24 | 0.55 |
| InternLM Dong et al. (2024) | 0.48 | 0.27 | 0.29 | 0.58 | 0.25 | 0.32 | 0.26 | 0.53 |
| InternLM-VL Dong et al. (2024) | 0.41 | 0.20 | 0.17 | 0.47 | 0.22 | 0.20 | 0.20 | 0.53 |
| InstructBLIP Dai et al. (2024) | 0.44 | 0.23 | 0.21 | 0.37 | 0.21 | 0.16 | 0.22 | 0.47 |
| SPHINX Lin et al. (2023) | 0.56 | 0.28 | 0.32 | 0.59 | 0.24 | 0.33 | 0.27 | 0.58 |

simple for humans, the VSP benchmark poses a substantial challenge to MLLMs, illustrating that current MLLMs have considerable potential for improvement in spatial planning tasks.

**MLLMs face significant difficulties with spatial planning in dynamic and realistic environments.** Based on the experimental results, most models perform worse when tested in dynamic (*Collision* scenario) and realistic (*Google Map* scenario) settings. We identify two major reasons for this: First, the tasks in the *Collision* scenario are complex and typically require multiple capabilities. For example, to assess collision danger, the model must locate both the player and the car on the map, calculate the time needed for the player to reach the goal, and determine if the car will hit the player during that time. This poses significant challenges for the models. Second, the inputs in the *Google Map* scenario are intricate and contain a series of irrelevant symbols, making it difficult for the model to accurately interpret the map. The models' performance in these environments suggests that current models are not yet equipped to handle spatial reasoning in such complex environments. Consequently, we focus on the *Maze Navigation* and *Blocks World* scenarios in the following experiments to diagnose the models' hidden weaknesses in VSP tasks.

**Quick performance decay as difficulty increases.** We observe a significant drop in the success rates of MLLMs as task difficulty escalates. For example, GPT-Vision may achieve a success rate of over 50% on 3x3 size maps, but this plummets to just 10% on 8x8 maps. Analyzing the impact of increased difficulty, we identify two major challenges for the models: First, increasing size of the map in *Maze Navigation* scenario could make it difficult for the model to accurately *perceive* the positions of elements within the map. Second, the increase in both map size and the number of steps required for moving blocks heightens the challenge for the model to *reason* deeply through the entire path and devise a complete, viable solution. In the following experiments, we focus on these two factors and provide in-depth analysis with subsequent tasks.

**Challenges in open-source models.** Finally, we note that open-source models often face challenges when evaluating on these tasks. We identify two main factors. ① Context length: Open-source models typically have significantly shorter context windows compared to private models. Besides, image embeddings can occupy many tokens. Thus, these models may not have enough capacity to understand the complete inputs. For example, LLAVA-V1.6-VICUNA-7B is trained with a maximum context window of 2048 tokens, while each image consumes 576 tokens. Consequently, when fed with multiple images and relatively long texts in our tasks, the total token length may surpass training, resulting in poor performance. ② Multiple image input: Our tasks require the model to understand multiple images interleaved with text inputs, whereas many open-source models are only trained with single-image inputs, with the image positioned at the start of the input. To further explore their potential in our tasks, we assess their performance after training on our inputs in Section 4.6. Meanwhile, we suggest that future open-source models could consider increasing their context length and reducing restrictions on input formats to address complex and realistic tasks effectively.

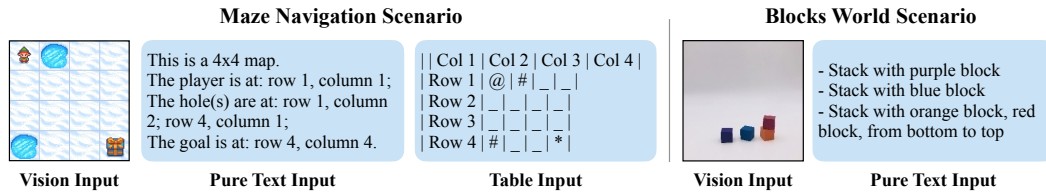

Figure 5: The visual and corresponding textual inputs.

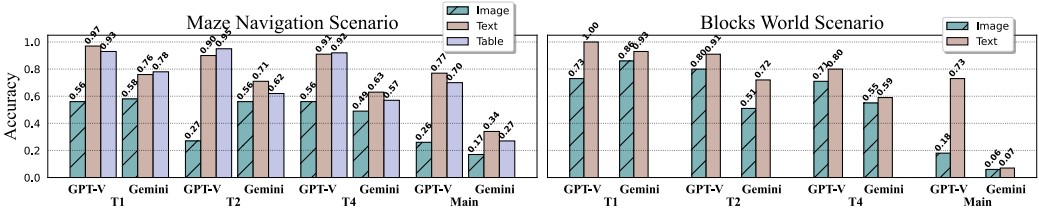

Figure 6: Performance comparison with the visual/textual input. When the environment is described by text instead of image, the performance increases significantly.

### 4.3 THE PERCEPTION AND REASONING SUB-TASKS EVALUATION

From the previous observation, we identify that spatial *perception* and *reasoning* could be two important capabilities for an agent to successfully perform visual spatial planning. Next, we evaluate these two abilities through the remaining tasks T1-T4. Similar to previous setting, all the evaluation is conducted under zero-shot settings.

Table 3 shows the decomposed capability results. GPT-4o and GPT-Vision perform well across many tasks, showing decent perception and reasoning capabilities. However, the overall performance of private models hovers around 50%, which is far from satisfactory for agents requiring spatial intelligence. Furthermore, the performance of open-source models is mostly close to random guessing on these tasks, indicating significant gaps compared to private models. One caveat is that while T4 focuses on reasoning capabilities, it still relies on the perception capabilities because the input still contains images. We perform further analysis to disentangle these two abilities in Section 4.4.

### 4.4 THE EFFECTS OF VISUAL INPUT PERCEPTION AND REASONING

Previous analysis shows that even current state-of-the-art models have clear deficiencies in various aspects of visual spatial planning. In this study, we focus on disentangling the effects of perception and reasoning by exploring the performance gain assuming the model had perfect perception.

The key strategy here is to create a scenario where the model has acquired all necessary information that would typically be obtained through visual perception. To this end, for every input image, we produce the corresponding textual inputs and replace those images, as shown in Figure 5. For the *Maze Navigation* scenario, we use either pure text descriptions or tables to depict the image. For the *Blocks World* scenario, we use pure text descriptions. We do not use tables for the *Blocks World* scenario because the number of blocks in each horizontal stack is usually unequal, making it difficult to form a complete table. Appendix B includes complete examples with pure text or table input.

The results are shown in Figure 6. We observe a clear performance improvement when using textual input across every task. This suggests image perception presents significant challenges for MLLMs, and poor perception ability is a key factor in the inferior performance observed in previous tasks. Meanwhile, we observe that even with textual input, Gemini still cannot achieve decent performance on tasks that require reasoning. This indicates deficiencies in its reasoning capabilities as well.

### 4.5 IN-CONTEXT LEARNING IN VISUAL SPATIAL PLANNING

In-context learning is a widely-adopted method to enhance LM's reasoning ability Brown et al. (2020). In this analysis, we study if it boosts the visual spatial planning capabilities. We included varying

Table 4: Effects of providing in-context examples.

| Task | MAZE NAVIGATION | | | | | BLOCKSWORD | | | | |
|---|---|---|---|---|---|---|---|---|---|---|
| | **T1** | **T2** | **T3** | **T4** | **Main** | **T1** | **T2** | **T3** | **T4** | **Main** |
| Gemini, 0-shot | 0.58 | 0.56 | 0.33 | 0.49 | 0.17 | 0.86 | 0.51 | 0.54 | 0.55 | 0.03 |
| Gemini, 1-shot | 0.50 | 0.66 | 0.31 | 0.48 | 0.20 | 0.91 | 0.68 | 0.71 | 0.59 | 0.03 |
| Gemini, 2-shot | 0.53 | 0.68 | 0.31 | 0.51 | 0.21 | 0.90 | 0.76 | 0.70 | 0.61 | 0.03 |
| Gemini, 4-shot | 0.53 | 0.67 | 0.35 | 0.53 | 0.19 | 0.91 | 0.64 | 0.69 | 0.62 | 0.06 |
| GPT-Vision, 0-shot | 0.56 | 0.27 | 0.46 | 0.56 | 0.26 | 0.73 | 0.80 | 0.70 | 0.71 | 0.10 |
| GPT-Vision, 1-shot | 0.55 | 0.50 | 0.47 | 0.57 | 0.28 | 0.89 | 0.84 | 0.94 | 0.73 | 0.11 |
| GPT-Vision, 2-shot | 0.55 | 0.63 | 0.50 | 0.56 | 0.30 | 0.90 | 0.83 | 0.95 | 0.71 | 0.16 |
| GPT-Vision, 4-shot | 0.54 | 0.69 | 0.54 | 0.56 | 0.29 | 0.90 | 0.79 | 0.96 | 0.73 | - |

Table 5: Fine-tuning results for open-source models.

| Model | Setting | MAZE NAVIGATION | | | | | BLOCKS OF WORLD | | | | |
|---|---|---|---|---|---|---|---|---|---|---|---|
| | | **T1** | **T2** | **T3** | **T4** | **Main** | **T1** | **T2** | **T3** | **T4** | **Main** |
| LLaVA | zero-shot | 0.49 | 0.27 | 0.21 | 0.54 | 0.05 | 0.22 | 0.21 | 0.24 | 0.55 | 0.01 |
| | fine-tune | 0.53 | 0.99 | 0.51 | 0.93 | 0.60 | 1.00 | 1.00 | 1.00 | 1.00 | 0.97 |
| InternLM | zero-shot | 0.48 | 0.27 | 0.29 | 0.58 | 0.11 | 0.25 | 0.32 | 0.26 | 0.53 | 0.00 |
| | fine-tune | 0.52 | 0.59 | 0.91 | 0.59 | 0.17 | 0.29 | 0.44 | 0.69 | 0.62 | 0.09 |

numbers of examples for Gemini and GPT-Vision (refer to Appendix C for the input examples). The result is shown in Table 4. There are two key observations: First, in-context examples make some potential contributions, but they are not significant. Introducing examples only benefits in several sparse cases, such as **T2** in *Maze Navigation* and **T3** in *Blocks World*. Second, scaling in-context examples generally does not help, as illustrated by the saturated performance in each task.

### 4.6 FINE-TUNING IN VSP TASKS

Finally, we assess the capabilities of the open-source model through dedicated training. We performed LoRA fine-tuning on `llava-v1.6-vicuna-7b` and `internlm-xcomposer2-7b`. The models are trained on 10k data points (image-text pairs) for each task and scenario. We use the default hyperparameters provided in the official repository. More fine-tuning details can be found in Appendix D. The results, shown in Table 5, demonstrate clear performance improvements for both models across a series of tasks, highlighting their potential in spatial planning. Additionally, we observe that LLaVA shows greater improvement compared to InternLM, suggesting that different model architectures may exhibit varying levels of efficacy in spatial planning capabilities.

## 5 CONCLUSION

We present VSP, a benchmark measuring and diagnosing the visual spatial planning capabilities in MLLMs. VSP quantifies the model's performance through a series of carefully designed tasks, with main tasks focusing on general spatial planning abilities and sub-tasks focusing on individual capabilities needed for the main task. Experiments show that both private and open-source models fail to generate effective plans for even simple spatial planning tasks, and further analyses expose their bottlenecks in spatial perception and reasoning abilities. Our work illuminates future directions for improving MLLMs' abilities in spatial planning.

## 6 REPRODUCIBILITY STATEMENT

Our dataset is anonymized and released on `https://anonymous.4open.science/r/Visual-Spatial-Planning-B131/README.md`. Besides, the input images and text for each task can be found in Appendix A. We also detail how the data are processed in Section 3.5 and Appendix A.

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
