# A  DETAILS OF BENCHMARKS

## A.1  ADDITIONAL TASK IMPLEMENTATION PROCESS AND STATISTICS

In Section 3.5, we described our general benchmark creation process. Our task creation process can be divided into three stages: (a) preparing the input image; (b) formulating task prompts; (c) developing scripts for auto evaluation. Additionally, some specific tasks require extra steps to implement, such as question and answer generation. In what follows, we describe these steps in detail.

**Maze Navigation, Main Task** In this task, agents need to find a safe path to navigate from the start grid to the goal. We adjust the map generation mechanisms to ensure the positions of the start grid, the goal, and the holes are all randomly generated, while ensuring there is at least one viable safe path from the start grid to the goal. For each grid, the probability that it contains the hole is 20%.

**Maze Navigation, T1** In this task, agents need to identify whether a specific grid is safe (i.e., whether it contains a hole or not). We randomly sample a row number and a column number and ask the safety question for this randomly chosen grid in each problem of this task. Additionally, to prevent the model from patterned guessing and achieving falsely high ratings (e.g., answering "not safe" for all images and obtaining high accuracy scores), we regenerate the map for this task to ensure that the safe and unsafe grids each comprise around 50% of the total grids in a single map.

**Maze Navigation, T3** In this task, agents need to find the correct textual description that fits the visual input. For each problem, we prepare four textual description candidates. One candidate is the correct answer, one has the correct size but an incorrect map arrangement, and the other two candidates have the wrong size. The candidates are shuffled to prevent the model from making random guesses.

**Maze Navigation, T4** In this task, agents need to determine if the given action series is safe or not. Similarly, to prevent the models from achieving falsely high ratings through guessing, we generate the action series to ensure that around 50% of them are safe and the other 50% are not. For the unsafe paths, the particular step in which the player steps into a hole is also randomly chosen.

**Blocks World, T2** In this task, agents need to determine the spatial relation between two designated blocks. In addition to the directional relation ("above" and "below"), we note that it is important for agents to recognize if two blocks are at the same stack or not. Therefore, we design the following four candidates for this question: (A) The first block is directly above the second block, and they are in the same stack; (B) The first block is directly below the second block, and they are in the same stack; (C) The two blocks are at different stacks; (D) At least one of the mentioned blocks do not exist in the presented image.

**Blocks World, T4** In this task, agents need to determine the consequence of a given moving plan. Specifically, some invalid moving plans contain actions that cannot be executed in the given scenario, such as trying to move a block that is covered by another block. To prevent guessing in this task, similar to the maze navigation scenario, we generate the action plans to ensure that half of the plan candidates are executable. For the plans that cannot be executed, there can be two types of errors: first, the plan may include steps that involve moving a block from or to an invalid position; second, the plan may try to move a block that does not exist. We randomly generate these errors in inputs.

**Google Map** For each task in this scenario, the input image is derived from real Google map intersections. To ensure that the roads on the map align with these four directions (north, south, east, west), we selected a map of New York City, where the streets and avenues intersect at 90 degrees. The map has been rotated to ensure that streets and avenues are strictly aligned in one of these cardinal directions. The starting location and the goal are randomly chosed crossroads in the map.

**Statistics** The VSP benchmark consists of 12 tasks in four scenarios. In *Maze Navigation* and *Blocks World* scenarios, the problems are designed with different difficulty levels. Each difficulty level consists of 100 problems. In total, the VSP benchmark includes 4.6k questions.

## A.2  COMPLETE PROMPT

In this subsection, we provide the complete prompts for each task. Generally, the prompts consist of a general task description at the beginning and a specific question at the end. The prompts interleave text and images in a pattern similar to a human-readable manual with reference figures.

**Prompt for Maze Navigation scenario, Main task (Spatial Planning):**

As a professional maze solver, your task is to analyze a grid-based map and devise an action plan that enables a player to reach the goal from the starting point without falling into any holes, using the fewest possible moves. Since coding is not within your skill set, your approach relies on logical reasoning of the map.

## Game Setup
- The game presents a fully observable grid-based map.
- The player starts at a specified grid square, with the goal located elsewhere on the map.
- Each grid square is either safe or contains a hole.
- Your goal is to guide the player to the goal while avoiding holes.
The following figure shows how the player, the holes (non-safe grid), the lands (safe grids), and the goals look like.

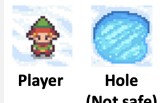 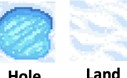 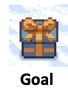

Player    Hole (Not safe)    Land (Safe)    Goal

## Moving Rules
- The action plan involves a series of moves: 'L' (left), 'R' (right), 'U' (up), or 'D' (down).
- Each move transfers the player to the adjacent square in that direction, provided it is a safe square. The player cannot move more than one square at a time.
- Moving off the edge of the map has no effect. The player will remain at the same square.
- DO NOT MOVE INTO A HOLE! Falling into a hole results in defeat.
- Locating at the grid containing the goal results in victory.
We provide an example to further illustrate the rules.

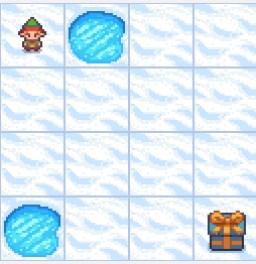

In this provided example:
- The player is at Row 1, Column 1;
- The goal is at Row 4, Column 4;
- There are two holes: one at Row 1, Column 2, and another at Row 4, Column 1.
- The player can move DOWN. This is because moving down brings them to Row 2, Column 1, and this cell is safe (without holes).
- Moving UP has no effects. This is because the player is already in the topmost row.
- Similarly, moving LEFT has no effects because the player is already in the left-most column.
- Moving RIGHT places the player at Row 1, Column 2. Since there is a hole at this grid, this move results in a loss.

## Procedure and Output
Now you will solve the given maze. To solve it, please generate text exactly follow the following steps:
1. First, interpret map. List where the player is at now, where is the goal, and where are the holes.
2. Then, generate an action plan to navigate to the goal step by step. At each step, you should check:
(a) Where the current move leads the player to (the row and column);
(b) What is in that grid. Is it a hole? Is it the goal? Is it an empty space?
(c) Determine if that is a safe action. If not, correct it and re-generate the action plan.
3. Next, verify if the steps successfully navigate the player to the goal without falling into the hole. If not, restart from step 2 and re-generate this step.
*(continue in next page)*

*(Continued)*

4. If succeed, output an aggregated plan using "Action plan: <PLAN>", where <PLAN> is a string concatenated action in each step. For example, "Action plan: L,L,R,U,D" meaning an action plan of left, left, right, up, and down. Double check the final action plan is consistent with the previous analysis.

Do not output any extra content after the above aggregated output.

Please generate the action plan for the following maze:

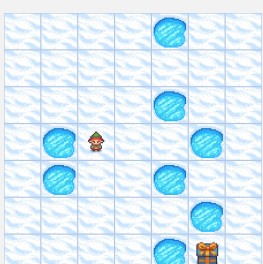

---

**Prompt for Maze Navigation scenario, Task 1 (Single Object Perception):**

In this task, you will analyze a maze to determine if there is a hole in a specific position.
The following figure illustrates the appearances of the player, holes, lands, and the goal within the maze. You will need to focus on the appearance of the hole.

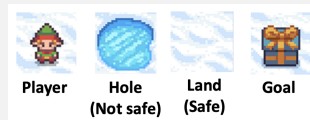

Here is an example to illustrate how to analyze and answer the question:

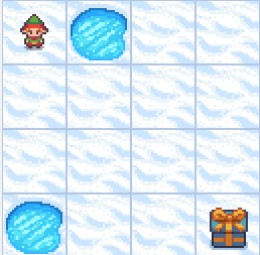

Example question: Is there a hole in row 3, column 3?

In this example:
- We check the position in row 3, column 3.
- According to the image, it is a land square. It does not contain a hole.
- Therefore, you will output "<Output> No".

Your output should be: "<Output> No" or "<Output> Yes", depending on whether there is a hole at the specified position.

Now you will analyze the following maze and answer the question:
Is there a hole in row 2, column 1?

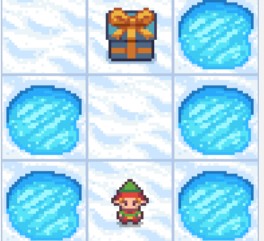

972
973
974
975
976
977
978
979
980
981
982
983
984
985
986
987
988
989
990
991
992
993
994
995
996
997
998
999
1000
1001
1002
1003
1004
1005
1006
1007
1008
1009
1010
1011
1012
1013
1014
1015
1016
1017
1018
1019
1020
1021
1022
1023
1024
1025

**Prompt for Maze Navigation scenario, Task 2 (Spatial Relation Perception):**

In this task, you will analyze a maze to determine the relative positions of the player and the goal.
The following figure illustrates the appearances of the player, holes, lands, and the goal within the maze. You will need to focus on the player and the goal.

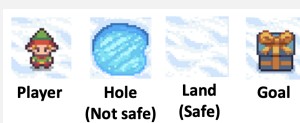

To describe their relative positions, use the directional indicators from "Above", "Below", "Left", "Right". We provide an example to illustrate how to interpret and describe these positions:

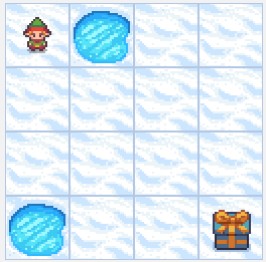

In this example:
- We focus on the position of the player and the goal.
- Rows: The player is at row 1, and the goal is at row 4. Here, the row number is from top to bottom. Comparing player (row=1) with goal (row=4), player is counted first. Therefore, the player is positioned above the target.
- Columns: The player is at column 1, and the goal is at column 4. Here, the column number is from left to right. Comparing player (column=1) with goal (column=4). Therefore, the player is to the left of the target.
- Remember that we should answer the player's position with respect to the goal, not the opposite. Therefore, we answer "Above,Left".

Your output should be two parts:
1. Analyze the rows and columns of the player and the goal like shown above.
2. Following your analysis, output answer as "<Output> <Position>". For example, "<Output> Above,Left" means the player is above and to the left of the goal, and "<Output> Below" means the player is below the goal. Note that you should not output "Left" or "Right" if the player and the goal are at the same column, and similarly, you should not output "Above" or "Below" if the player and the goal are at the same row.

Now you will analyze the following maze and determine the relative position of the player in relation to the goal.

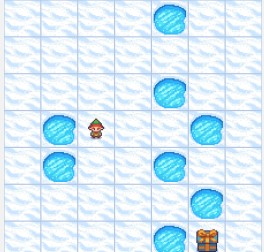

**Prompt for Maze Navigation scenario, Task 3 (Environment Perception):**

In this task, you will analyze a maze presented in an image. Later, you will be presented with four choices, each offering a textual representation of a candidate maze. You will need to choose the representation that exactly reflects the contents of the given image.
The following figure illustrates the appearances of the player, holes, lands, and the goal within the maze in the image.

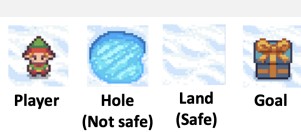

This is how the player, the holes (non-safe grid), the lands (safe grids), and the goals look like in a map:
- The player is represented as "@"
- The hole is represented as "#"
- The safe grid is represented as "_"
- The goal is represented as "*"
- If the player is at the goal (at this case the game is solved), that grid is represented as "%"

We provide an example to illustrate how to interpret the input, candidates, and answer the question. Here is the image input:

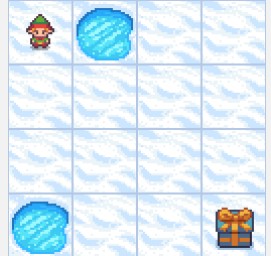

Here are the textual candidates:
(A)
```
| | Col 1 | Col 2 | Col 3 |
| Row 1 | # | _ | _ |
| Row 2 | # | @ | # |
| Row 3 | _ | * | _ |
```

(B)
```
| | Col 1 | Col 2 | Col 3 | Col 4 | Col 5 |
| Row 1 | _ | _ | _ | _ | _ |
| Row 2 | _ | # | _ | _ | _ |
| Row 3 | _ | # | * | _ | # |
| Row 4 | _ | @ | _ | _ | _ |
| Row 5 | _ | _ | _ | # | _ |
```

(C)
```
| | Col 1 | Col 2 | Col 3 | Col 4 |
| Row 1 | @ | # | _ | _ |
| Row 2 | _ | _ | _ | _ |
| Row 3 | _ | _ | _ | _ |
| Row 4 | # | _ | _ | * |
```

(D)
```
| | Col 1 | Col 2 | Col 3 | Col 4 |
| Row 1 | _ | _ | _ | _ |
| Row 2 | * | _ | _ | _ |
| Row 3 | @ | _ | # | _ |
| Row 4 | _ | _ | _ | # |
```

Here is an example of how to analyze and answer the question:
*(continue in next page)*

*(Continued)*
- First, we focus on the difference of the maze shape between the candidates and the input image.
- We begin by examining the input image. It is a 4-by-4 maze. We then review the candidates. Candidate A is a 3-by-3 maze. Therefore, it is not the correct answer. Similarly, Candidate B is a 5-by-5 maze, which also cannot be correct. Both Candidate C and Candidate D are 4-by-4 mazes. Now we only need to choose from them.
- For the remaining candidates, we compare the positions of the players, goals, and the holes in the maze.
- We first check the input image. What is the position of the player in the image? The player is in row 1, column 1. We then check the remaining candidates. For Candidate C, the textual representation indicates the player is also at row 1, column 1, matching the input image. For Candidate D, the player is located at row 3, column 1. Hence, Candidate D is not the correct answer.
- We double check the remaining Candidate C, and it correctly shows the position of the player, holes, and the goal. It is therefore the correct answer.
<Answer> C

Your output should consist of two parts:
1. First, analysis the input image and candidates similar to the reasoning process above.
2. Following the reasoning process, output answer as "<Answer> <Choice>", where "<Choice>" is one of A,B,C,D.
Important: Note that there will be only one correct answer. If you find no answer or multiple answers, you must go back and recheck your reasoning process. You are not allowed to provide 0 or more than 1 answer.

Now answer the question below. Here is the image input:

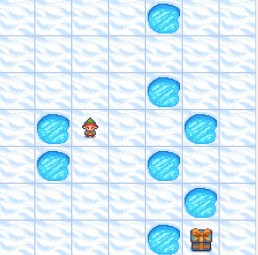

Here are the textual candidates:

<CANDIDATES>

---

**Prompt for Maze Navigation scenario, Task 4 (Spatial Reasoning):**

You are a maze-solving agent playing a pixelated maze video game.
Mazes are presented on grid maps, where each tile can be empty land, or contain a player, hole, or goal.
Each of the above tile types are represented as square pixel art images.

In this task, you will analyze a grid-based map and determine if a provided action plan is safe. A safe action plan avoids stepping into holes in the map.
The following figure illustrates the appearances of the player, holes, lands, and the goal within the maze.

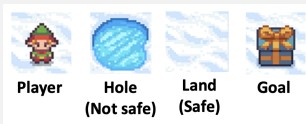

## Moving Rules
- The action plan involves a series of moves: 'L' (left), 'R' (right), 'U' (up), or 'D' (down).
- Each move transfers the player to the adjacent square in that direction, provided it is a safe square. The player cannot move more than one square at a time.
- Moving off the edge of the map has no effect. The player will remain at the same square.
- DO NOT MOVE INTO A HOLE! Falling into a hole results in defeat.
- Locating at the grid containing the goal results in victory.
We provide an example to further illustrate the rules.
*(continue in next page)*

*((Continue))*

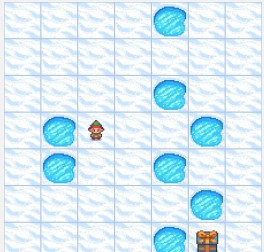

In this provided example:
- The player is at Row 1, Column 1;
- The goal is at Row 4, Column 4;
- There are two holes: one at Row 1, Column 2, and another at Row 4, Column 1.
- The player can move DOWN. This is because moving down brings them to Row 2, Column 1, and this cell is safe (without holes).
- Moving UP has no effects. This is because the player is already in the topmost row.
- Similarly, moving LEFT has no effects because the player is already in the left-most column.
- Moving RIGHT places the player at Row 1, Column 2. Since there is a hole at this grid, this move results in a loss.

## Procedure and Output
Your output should include the following parts:
1. First, interpret map. List where the player is at now, where is the goal, and where are the holes.
2. Then, reasoning by following the given action plan. At each step, you should check:
(a) Where the current move leads the player to (the row and column);
(b) What is in that grid. Is it a hole? Is it the goal? Is it an empty space?
(c) Determine if that is a safe action.
3. Output if the action sequence is safe using "<Output> Yes" or "<Output> No". A safe action sequence should not include any unsafe actions.

Now please determine if the action sequence is safe for this given maze:

The action sequence is:
Up, Down, Right, Down, Right

**Prompt for Block World scenario, Main Task (Spatial Planning):**

You are a robot that sorts and organizes colored blocks by adding and removing them to stacks. You can move them between stacks to produce a desired end state.

In this task, you will see two photos of blocks. These photos show the beginning and end state of the blocks. Your task is to find a shortest movement plan to transit from the beginning state to the end state. Since coding is not within your skill set, your approach relies on logical reasoning of the map.

## Game Setup
- The stacks of blocks are presented in images. You must view and interpret the image in order to determine which blocks are in which stack and determine how to move them.
- Each block has a unique color (blue, yellow, purple, orange, red, green).
- Blocks are stacked vertically in a stack, forming multiple stacks. All stacks are on the table.
- In a single move, you can only move the top block of any pile. Attempting to move lower blocks is considered an invalid move.
- You can either (a) move the top block to the top of another stack, or (b) place the top block on the table, creating a new stack with just one block.

We provide an example to further illustrate the rules:

This example features four blocks arranged in three stacks:
- Stack 1: Purple block (alone)
- Stack 2: Blue block (alone)
- Stack 3: From bottom to top: Orange block, Red block
You can only move the top block of each stack: the purple block, the blue block, and the red block. The orange block is stuck underneath the red block and cannot be moved directly.
Each move can place the block on another stack or on the table (creating a new stack of one). For instance, you could move the red block to either the blue stack or the table.
**Important Note**: The order of the stacks doesn't matter in this game. Two images are considered equivalent as long as the stacks contain the same blocks, regardless of the order in which the stacks appear. For example, an image with stack A on the left and stack B on the right is equivalent to an image with stack B on the left and stack A on the right.

## Procedure and Output
Your output should follow this format:
1. First, analyze the starting and ending configurations, including the number of stacks and the blocks in each stack (similar to the example above).
2. Then, list the moves in a step-by-step manner using the format move(SOURCE, TARGET). Remember, "SOURCE" refers to the block being moved (always the top block of a stack), and "TARGET" refers to the destination (another stack or the table).
## Example Output
<Analysis>
Starting state: there are three stacks:
- Stack 1: Purple block (alone)
- Stack 2: Blue block (alone)
- Stack 3: From bottom to top: Orange block, Red block
Ending state: there are three stacks:
- Stack 1: Purple block (alone)
- Stack 2: From bottom to top: Orange block, Blue block
- Stack 3: Red block (alone)
<Output>
1. move(red,table)
2. move(blue,orange)

*(continue in next page)*

*(Continue)*
Now please generate moving plan. The beginning state is:

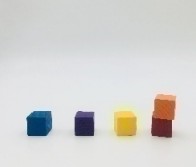

The end state is:

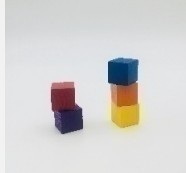

---

**Prompt for Block World scenario, Task 1 (Single Object Perception):**

In this task, you will see a photo of blocks. You will analyze the block configuration and then answer a question regarding the color of blocks in a specific place.

## Game Setup
- Each block has a unique color (blue, yellow, purple, orange, red, green).
- Blocks are stacked vertically in a stack, forming multiple stacks.
- In the questions, the position of the blocks is represented as "Stack s, Level l". The stack number is counted from left to right, and the level number is counted from bottom to top.
We provide an example to further illustrate the setting:

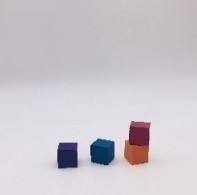

In this example, there are four blocks in three stacks. From left to right:
- Stack 1 has one level. Level 1 contains a purple block.
- Stack 2 has one level. Level 1 contains a blue block.
- Stack 3 has one level. From bottom to top: level 1 has an orange block, and level 2 has a red block.
As such, for the question "What is the color of the block at stack 3, level 1?", the correct answer is "<Output> orange".

## Procedure and Output
Your output should follow this format:
1. First, analyze the block configuration;
2. Then, answer the question with the format <Output> <Color>, where <Color> is one of (blue, yellow, purple, orange, red, green). For example, "<Output> red".

Now please answer the following question based on the given image below:
What is the color of the block at stack 1, level 2?

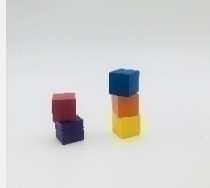

**Prompt for Block World scenario, Task 2 (Spatial Relation Perception):**

In this task, you will see a photo of blocks. You will analyze the block configuration and then answer a question regarding the spatial relation of two specified blocks.

## Game Setup
- Each block has a unique color (blue, yellow, purple, orange, red, green).
- Blocks are stacked vertically in a stack, forming multiple stacks.
- The possible relations of two blocks include: (A) One block is directly above another block, and they are at the same stack; (B) One block is directly below another block, and they are at the same stack; (C) Two blocks are at different blocks; (D) At least one of the asked blocks does not exist in the image.

We provide an example to further illustrate the rules:

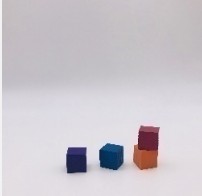

In this example, there are four blocks in three stacks. From left to right:
- Stack 1: Purple block (alone)
- Stack 2: Blue block (alone)
- Stack 3: From bottom to top: Orange block, Red block
We can answer the question regarding the spatial relations accordingly. For example, for the question "What is the spatial relation between the purple block and blue block?", since the purple block and the blue block are at different stacks, we should choose the choice indicating they are at different stacks.

## Procedure and Output
Your output should follow this format:
1. First, analyze the block configuration;
2. Then, answer the question with the format <Output> <Choice>, where <Choice> is one of A,B,C,D. For example, "<Output> A".

Now please answer the following question based on the given image below:
What is the spatial relation between the red block and the yellow block?
(A) The red block is directly above the yellow block, and they are in the same stack.
(B) The red block is directly below the yellow block, and they are in the same stack.
(C) The red block and the yellow block are at different stacks.
(D) At least one of the red block and the yellow block does not exist in the presented configurations.

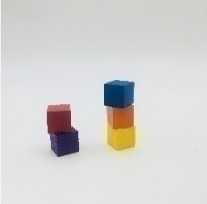

**Prompt for Block World scenario, Task 3 (Environment Perception):**

In this task, you will analyze an image containing several stacks of blocks. Later, you will be presented with four choices, each offering a textual representation of a block configuration. You will need to choose the configuration that exactly reflects the contents of the given image.
## Game Setup
- Each block has a unique color (blue, yellow, purple, orange, red, green).
- Blocks are stacked vertically in a stack, forming multiple stacks.
This is an image input example:

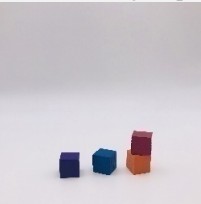

This example features four blocks arranged in three stacks:
- Stack 1: Purple block (alone)
- Stack 2: Blue block (alone)
- Stack 3: From bottom to top: Orange block, Red block
Here are examples of textual representations:
(A)
- Stack with red block, yellow block, from bottom to top
- Stack with orange block, purple block, green block, from bottom to top

(B)
- Stack with purple block
- Stack with blue block
- Stack with orange block, red block, from bottom to top

(C)
- Stack with orange block
- Stack with purple block
- Stack with blue block
- Stack with green block, yellow block, from bottom to top

(D)
- Stack with green block
- Stack with yellow block, blue block, from bottom to top
- Stack with red block, orange block, from bottom to top

We can analyze which text representation exactly reflects the configurations in the image accordingly. In this example:
- The input image has 3 stacks, while Candidate A only has 2 stacks. Therefore, Candidate A is not the correct answer.
- Similarly, Candidate C has 4 stacks, which also cannot be correct.
- For Candidate B, the blocks in each stack match what's shown in the image. This is the correct answer.
- For Candidate D, the blocks in each stack do not match the image. For example, stack 1 in the image has a purple block, and there is no any purple block in Candidate D. So this is incorrect.
- Therefore, the final answer is B.
## Procedure and Output
Your output should follow this format:
1. First, analyze the block configuration in the image and candidates as shown above;
2. Then, answer the question with the format <Output> <Choice>, where <Choice> is one of A,B,C,D. For example, "<Output> A".

Now please choose the correct textual representation based on the given image below:

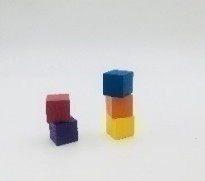

Here are the textual candidates:
<CANDIDATES>

**Prompt for Maze Navigation scenario, Task 4 (Spatial Reasoning):**

You are a robot that sorts and organizes colored blocks by adding and removing them to stacks. You can move them between stacks to produce a desired end state.

In this task, you will see a photo of blocks. This photo shows the beginning state of the blocks. You will see a photo of blocks. This photo shows the beginning state of the blocks. Meanwhile, you will be provided an action sequence about moving blocks. Your task is to determine if the provided action plan can be successfully executed.

## Game Setup

- The block configuration is presented in the image. You must view and interpret the image in order to determine which blocks are in which stack and determine the consequence of moving.
- Each block has a unique color (blue, yellow, purple, orange, red, green).
- Blocks are stacked vertically in a stack, forming multiple stacks.
- A valid action can only move the top block of any stacks. Attempting to move lower blocks is considered an invalid move.
- For the destination, a valid move can either (a) move the top block to the top of another stack, or (b) place the top block on the table, creating a new stack with just one block.

We provide an example to further illustrate the rules:

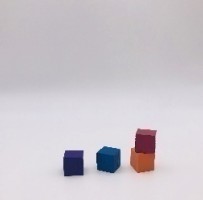

The sequence of actions provided is: 1. move(red,table) 2. move(green,table)

In this example, there are four blocks in three stacks. The stacks are:
- Stack 1: Purple block (alone)
- Stack 2: Blue block (alone)
- Stack 3: From bottom to top: Orange block, Red block

It is valid to move the purple block, the blue block, and the red block, since they are at the top of a stack. It is invalid to move the orange block since it is not at the top of a stack (because it is covered by the red block). Each move can place the block on top of another stack or on the table (creating a new stack of one). For instance, you could move the red block to either the blue stack or the table.

## Procedure and Output

Your output should follow this format:

1. First, briefly analyze the block configuration, and check each action step by step to see if the provided step is valid as shown above.
2. Then, answer the question with the format "<Output> Yes" or "<Output> No" to indicate if the action sequence is valid.

Here is an example for the output:

<Analysis> In the image, there are three stacks:
- Stack 1: Purple block (alone)
- Stack 2: Blue block (alone)
- Stack 3: From bottom to top: Orange block, Red block

The first action "move(red,table)" is valid, because the red block is on top of a stack (stack 3 in this case), and the target is "table". After the first action, the state will become:
- Stack 1: Purple block (alone)
- Stack 2: Blue block (alone)
- Stack 3: Orange block (alone)
- Stack 4: Red block (alone)

The second action "move(green,table)" is invalid, because there is no green block.

Therefore, the provided action sequence is invalid.

<Output> No

Now please determine if the provided action sequence is valid given the following input state:

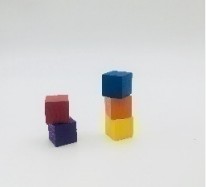

The action sequence is: 1. move(red,table) 2. move(yellow,red) 3. move(purple,table)

**Prompt for Google Map scenario:**

As a professional pathfinder, your task is to analyze a map and find a route from the starting location to the goal.

## Game Setup
- The game presents a fully observable map.
- The starting location is marked with blue "S", and the goal is marked with red "G".
- Your goal is to find a path from the starting location to the goal.

## Moving Rules
- The action plan involves moves in four directions: 'W' (West), 'E' (East), 'N' (North), or 'S' (South).
- Each move is along with distances. Distances are measured by how many crossroads passed.
We provide an example to further illustrate the rules.

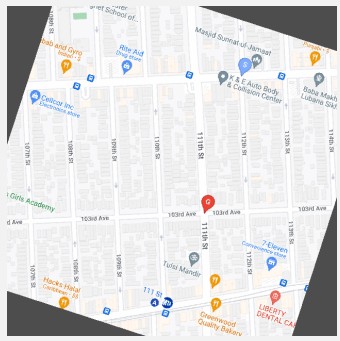

In this provided example:
- You are now at the southwest of the goal.
- If you move north by 1 crossroad, you will be at the west of the goal.
- If you move east by 4 crossroads, you will be at the goal.
- IMPORTANT: Please ignore the name of the street and avenue. The numbers in the name cannot be used to compute how many crossroads need to be passed.

## Procedure and Output
Now you will solve the given maze. To solve it, please first analyze the relative spatial relation between the starting point and the goal (for example, southwest). Then, output a path using the format <Direction>: <Number of crossroads passed>.
For example:
<Output>
1. North: 1
2. East: 4
means move north by 1 crossroad, and move east by 4 crossroads.
<Output>
1. South: 1
means move south by 1 crossroad.
Do not output any extra content after the above aggregated output.

Please output path for the following map:

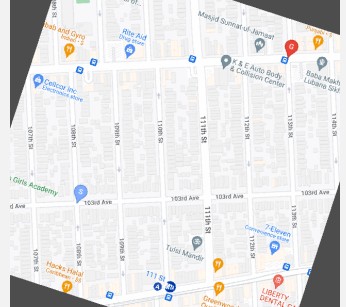

**Prompt for Collision scenario:**

As a professional navigation agent, your task is to analyze a map and determine the time needed for the car and the person passing the goal. Then, determine if there is a danger of collision.

## Game Setup
- The game presents a fully observable map. There is a person, a car, and a goal on the map.
- The game further specifies the moving direction of the person and car ("up", "down", "left", "right").
- Your will need to determine the time needed for the car and the person passing the goal through calculation.
- If the car and the person reaches the goal within 0.5 second, then there is a danger of collision.
The following figure shows how the person, the car, and the goals look like.

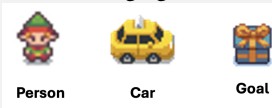

We provide an example to further illustrate the rules.

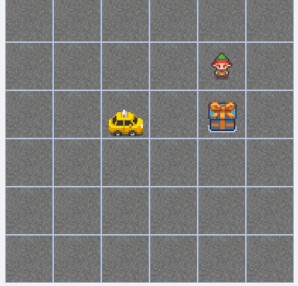

The car is moving right with speed 1.0 grid per second, and the person is moving down with speed 0.5 grid per second.

In this provided example:
- The car is 2 grid away from the goal. Given it's time as 1.0 grid per second, the time needed is 2 / 1.0 = 2 seconds.
- The person is 1 grid away from the goal. Given it's time as 0.5 grid per second, the time needed is 1 / 0.5 = 2 seconds.
- Since they both reach the goal in 2 seconds, there is a danger of collision.

## Procedure and Output
Now you will answer for the following given map. To solve it, analyze the car and the person separately. Then, answer for them separately. For example:
Car: 2.0
Person: 2.0
Yes
means car and the person will need 2.0 seconds to pass the goal respectively, and there is a danger of collision. Do not output any extra content after the above aggregated output.

Now please analyze and determine the time needed for the car and the person passing the goal:

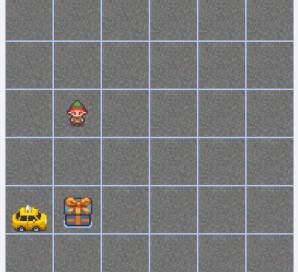

The car is moving right with speed 2.0 grid per second, and the person is moving down with speed 1.0 grid per second.

# B  PROMPT FOR TEXTUAL INPUT

In Section 4.4, we described the procedure of using textual representation instead of visual input. Below, we use the main task in the maze navigation scenario as an example to show the complete prompt after making the replacement.

---

**Prompt for Maze Navigation scenario, Main task (Spatial Planning, Textual Input):**

As a professional maze solver, your task is to analyze a grid-based map and devise an action plan that enables a player to reach the goal from the starting point without falling into any holes, using the fewest possible moves. Since coding is not within your skill set, your approach relies on logical reasoning of the map.

## Game Setup
- The game presents a fully observable grid-based map.
- The player starts at a specified grid square, with the goal located elsewhere on the map.
- Each grid square is either safe or contains a hole.
- Your goal is to guide the player to the goal while avoiding holes.

## Moving Rules
- The action plan involves a series of moves: 'L' (left), 'R' (right), 'U' (up), or 'D' (down).
- Each move transfers the player to the adjacent square in that direction, provided it is a safe square. The player cannot move more than one square at a time.
- Moving off the edge of the map has no effect. The player will remain at the same square.
- DO NOT MOVE INTO A HOLE! Falling into a hole results in defeat.
- Locating at the grid containing the goal results in victory.

We provide an example to further illustrate the rules.
Example Input:
This is a 4x4 map.
The player is at: row 1, column 1;
The hole(s) are at: row 1, column 2; row 4, column 1;
The goal is at: row 4, column 4.

In this provided example:
- The player is at Row 1, Column 1;
- The goal is at Row 4, Column 4;
- There are two holes: one at Row 1, Column 2, and another at Row 4, Column 1.
- The player can move DOWN. This is because moving down brings them to Row 2, Column 1, and this cell is safe (without holes).
- Moving UP has no effects. This is because the player is already in the topmost row.
- Similarly, moving LEFT has no effects because the player is already in the left-most column.
- Moving RIGHT places the player at Row 1, Column 2. Since there is a hole at this grid, this move results in a loss.
## Procedure and Output
Now you will solve the given maze. To solve it, please generate text exactly follow the following steps:
1. First, interpret map. List where the player is at now, where is the goal, and where are the holes.
2. Then, generate an action plan to navigate to the goal step by step. At each step, you should check:
(a) Where the current move leads the player to (the row and column);
(b) What is in that grid. Is it a hole? Is it the goal? Is it an empty space?
(c) Determine if that is a safe action. If not, correct it and re-generate the action plan.
3. Next, verify if the steps successfully navigate the player to the goal without falling into the hole. If not, restart from step 2 and re-generate this step.
4. If succeed, output an aggregated plan using "Action plan: <PLAN>", where <PLAN> is a string concatenated action in each step. For example, "Action plan: L,L,R,U,D" meaning an action plan of left, left, right, up, and down. Double check the final action plan is consistent with the previous analysis.
Do not output any extra content after the above aggregated output.

Please generate the action plan for the following maze:
This is a 3x3 map.
The player is at: row 3, column 2;
There is no holes in this map;
The goal is at: Row 1, Column 2.

---

1620
1621
1622
1623
1624
1625
1626
1627
1628
1629
1630
1631
1632
1633
1634
1635
1636
1637
1638
1639
1640
1641
1642
1643
1644
1645
1646
1647
1648
1649
1650
1651
1652
1653
1654
1655
1656
1657
1658
1659
1660
1661
1662
1663
1664
1665
1666
1667
1668
1669
1670
1671
1672
1673

# C  PROMPT WITH IN-CONTEXT EXAMPLES

In Section 4.4, we described the procedure of including in-context example in the test. Below, we use the main task in the maze navigation scenario as an example to show the complete prompt after adding in-context examples.

---

**Prompt for Maze Navigation scenario, Main task (Spatial Planning, in-context examples):**

As a professional maze solver, your task is to analyze a grid-based map and devise an action plan that enables a player to reach the goal from the starting point without falling into any holes, using the fewest possible moves. Since coding is not within your skill set, your approach relies on logical reasoning of the map.

## Game Setup
- The game presents a fully observable grid-based map.
- The player starts at a specified grid square, with the goal located elsewhere on the map.
- Each grid square is either safe or contains a hole.
- Your goal is to guide the player to the goal while avoiding holes.
The following figure shows how the player, the holes (non-safe grid), the lands (safe grids), and the goals look like.

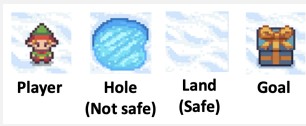

## Moving Rules
- The action plan involves a series of moves: 'L' (left), 'R' (right), 'U' (up), or 'D' (down).
- Each move transfers the player to the adjacent square in that direction, provided it is a safe square. The player cannot move more than one square at a time.
- Moving off the edge of the map has no effect. The player will remain at the same square.
- DO NOT MOVE INTO A HOLE! Falling into a hole results in defeat.
- Locating at the grid containing the goal results in victory.
We provide an example to further illustrate the rules.

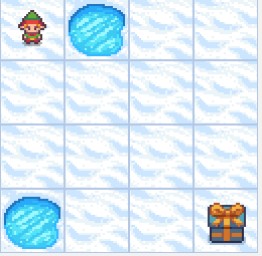

In this provided example:
- The player is at Row 1, Column 1;
- The goal is at Row 4, Column 4;
- There are two holes: one at Row 1, Column 2, and another at Row 4, Column 1.
- The player can move DOWN. This is because moving down brings them to Row 2, Column 1, and this cell is safe (without holes).
- Moving UP has no effects. This is because the player is already in the topmost row.
- Similarly, moving LEFT has no effects because the player is already in the left-most column.
- Moving RIGHT places the player at Row 1, Column 2. Since there is a hole at this grid, this move results in a loss.
*(continue in next page)*

---

*(Continue)*
## Procedure and Output
Now you will solve the given maze. To solve it, please generate text exactly follow the following steps:
1. First, interpret map. List where the player is at now, where is the goal, and where are the holes.
2. Then, generate an action plan to navigate to the goal step by step. At each step, you should check:
(a) Where the current move leads the player to (the row and column);
(b) What is in that grid. Is it a hole? Is it the goal? Is it an empty space?
(c) Determine if that is a safe action. If not, correct it and re-generate the action plan.
3. Next, verify if the steps successfully navigate the player to the goal without falling into the hole. If not, restart from step 2 and re-generate this step.

## Example:

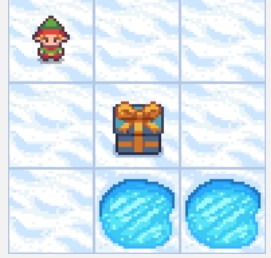

<Interpret>
The player is at row 1, column 1, and the goal is at row 2, column 2.
There are 2 holes. They are at: row 3, column 2; row 3, column 3.
<Action Plan>
- Moving Right (R). The player is now at row 1, column 2. This grid is safe.
- Moving Down (D). The player is now at row 2, column 2. This grid is the goal, so we stop here.
<Verification>
1. Right to row 1, column 2 (safe)
2. Down to row 2, column 2 (goal)
<Output>
Action plan: R,D

Please generate the action plan for the following maze:

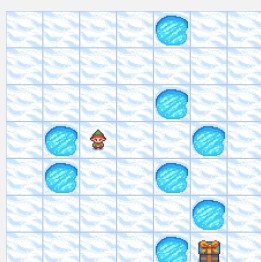

Table 6: Training details on LLaVA and InternLM-XComposer.

| | | Value |
|---|---|---|
| **LLaVA** | Learning rate | 2e-4 |
| | Scheduler | Cosine |
| | Epoch | 1 |
| | Training data | 10k |
| | Batch size | 32 |
| | Pretrained Checkpoint | `llava-v1.6-vicuna-7b` |
| **InternLM** | Learning rate | 5e-5 |
| | Scheduler | Cosine |
| | Epoch | 1 |
| | Training data | 10k |
| | Batch size | 8 |
| | Pretrained Checkpoint | `internlm-xcomposer2-7b` |

Table 7: Model performance on task 1 - 4 with different difficulties.

| Task 1 | MAZE NAVIGATION | | | | | | BLOCKSWORD | | |
|---|---|---|---|---|---|---|---|---|---|
| Difficulty | 3 | 4 | 5 | 6 | 7 | 8 | 3 | 4 | 5 |
| Gemini Team et al. (2023) | 0.63 | 0.58 | 0.61 | 0.45 | 0.64 | 0.54 | 0.86 | 0.82 | 0.89 |
| GPT-Vision Achiam et al. (2023) | 0.60 | 0.56 | 0.56 | 0.47 | 0.62 | 0.54 | 0.77 | 0.70 | 0.72 |
| Claude-3 AI (2024a) | 0.44 | 0.41 | 0.39 | 0.42 | 0.55 | 0.49 | 0.52 | 0.40 | 0.36 |
| GPT-4o AI | 0.72 | 0.65 | 0.57 | 0.44 | 0.56 | 0.53 | 0.98 | 0.94 | 0.92 |

| Task 2 | MAZE NAVIGATION | | | | | | BLOCKSWORD | | |
|---|---|---|---|---|---|---|---|---|---|
| Difficulty | 3 | 4 | 5 | 6 | 7 | 8 | 3 | 4 | 5 |
| Gemini Team et al. (2023) | 0.69 | 0.65 | 0.53 | 0.46 | 0.54 | 0.47 | 0.63 | 0.57 | 0.32 |
| GPT-Vision Achiam et al. (2023) | 0.37 | 0.27 | 0.22 | 0.30 | 0.29 | 0.18 | 0.86 | 0.77 | 0.77 |
| Claude-3 AI (2024a) | 0.65 | 0.65 | 0.70 | 0.65 | 0.67 | 0.70 | 0.59 | 0.57 | 0.43 |
| GPT-4o AI | 0.80 | 0.63 | 0.65 | 0.64 | 0.66 | 0.64 | 0.90 | 0.92 | 0.87 |

| Task 3 | MAZE NAVIGATION | | | | | | BLOCKSWORD | | |
|---|---|---|---|---|---|---|---|---|---|
| Difficulty | 3 | 4 | 5 | 6 | 7 | 8 | 3 | 4 | 5 |
| Gemini Team et al. (2023) | 0.38 | 0.32 | 0.36 | 0.23 | 0.37 | 0.30 | 0.62 | 0.51 | 0.49 |
| GPT-Vision Achiam et al. (2023) | 0.79 | 0.41 | 0.43 | 0.37 | 0.33 | 0.40 | 0.68 | 0.76 | 0.67 |
| Claude-3 AI (2024a) | 0.35 | 0.22 | 0.18 | 0.32 | 0.35 | 0.47 | 0.52 | 0.45 | 0.49 |
| GPT-4o AI | 0.89 | 0.72 | 0.49 | 0.4 | 0.39 | 0.59 | 0.85 | 0.95 | 0.90 |

| Task 4 | MAZE NAVIGATION | | | | | | BLOCKSWORD | | |
|---|---|---|---|---|---|---|---|---|---|
| Difficulty | 1 | 3 | 5 | 7 | 9 | 1 | 3 | 5 | 7 |
| Gemini Team et al. (2023) | 0.47 | 0.47 | 0.56 | 0.49 | 0.46 | 0.64 | 0.57 | 0.50 | 0.50 |
| GPT-Vision Achiam et al. (2023) | 0.62 | 0.55 | 0.57 | 0.52 | 0.53 | 0.66 | 0.70 | 0.74 | 0.73 |
| Claude-3 AI (2024a) | 0.57 | 0.60 | 0.60 | 0.59 | 0.67 | 0.72 | 0.65 | 0.60 | 0.68 |
| GPT-4o AI | 0.72 | 0.78 | 0.79 | 0.67 | 0.76 | 0.92 | 0.73 | 0.70 | 0.68 |

## D    TRAINING DETAILS

In this section, we describe the training details when we fine-tune LLaVA and InternLM-XComposer for our designed tasks. We perform LoRA fine-tuning, and we stick with the default hyperparameter settings in their official repo. The detailed hyperparameter choices are shown in Table 6.

## E    COMPLETE TASK PERFORMANCE RESULTS WITH DIFFERENT DIFFICULTY LEVELS

In this section, we present the experimental results of models across different difficulty levels. The results are shown in Table 7. As expected, we observe that as difficulty increases, all models perform progressively worse, with some performing close to random guessing at higher difficulty levels (*e.g.*, Task 1 in Maze Navigation scenario). We also observe that GPT-4o, the most recently released model, performs the best across different tasks, although it still frequently makes mistakes under different difficulty levels. This suggests a current bottleneck in state-of-the-art MLLMs.