# OpenReview forum: "VSP: Assessing the dual challenges of perception and reasoning in spatial planning tasks for MLLMs"
_ICLR.cc/2025/Conference — ICLR 2025 Conference Withdrawn Submission_

### Official Review · Reviewer_nsRH · 2024-10-31

**Soundness:** 2
**Presentation:** 3
**Contribution:** 2
**Rating:** 3
**Confidence:** 3

**Summary:**

This paper introduces a benchmark called Visual Spatial Planning (VSP) designed to evaluate the capabilities of multimodal large language models (MLLMs) in visual spatial planning. This includes understanding spatial arrangements of objects and devising action plans to achieve desired outcomes in visual scenes. The VSP benchmark comprises various scenarios, including Maze Navigation, Blocks World, Collision, and Google Map scenarios, each with different levels of complexity. The benchmark not only tests end-to-end spatial planning but also breaks down the task into finer-grained sub-tasks such as perception and reasoning. The evaluation results show that both open-source and private MLLMs struggle with even simple spatial planning tasks, revealing deficiencies in visual perception and reasoning abilities.

**Strengths:**

1. The benchmark presented in the paper tests the spatial planning capabilities of MLLMs through four different tasks, revealing their shortcomings.
2. The experiments are well-established, encompassing many of the MLLMs currently available in open and closed source, and also demonstrate performance improvements through fine-tuning.
3. The paper is generally well-written. figures can clearly reflect the setting of the tasks.

**Weaknesses:**

1. The proposed four tasks seem to have some gap with our real life. In real life, similar tasks (e.g., navigation) can currently be obtained by direct planning from maps without the need for planning by MLLM. In Collision Sceneario, speed and direction determination can be done by similar specialized algorithms. In other words, if MLLMs perform well on this benchmark, does it mean that MLLMs can demonstrate “spatial perception” in real life? The paper would have been more convincing if the authors could explain the necessity of these tasks for real-life applications.
2. From a psychological and physiological point of view, humans have the ability to perceive space to a certain extent dependent on the parallax brought about by binocular vision and the temporal difference brought about by the movement of objects. However, benchmark is more of a single RGB image and a task on 2D.
3. Depth is not introduced in benchmark. Depth is very important in spatial perception, and can directly reflect the distance of different objects and thus the distribution of objects. I think authors should discuss and compare this, like Spatialbot[1].
4. If the authors could provide some examples between benchmark and real applications, I think benchmark would be more contributing.

[1]SpatialBot: Precise Spatial Understanding with Vision Language Models

**Questions:**

see weaknesses

---

### Official Review · Reviewer_3Mmr · 2024-11-03

**Soundness:** 3
**Presentation:** 3
**Contribution:** 2
**Rating:** 5
**Confidence:** 4

**Summary:**

This paper introduces VSP (Visual Spatial Planning), a benchmark designed to evaluate the spatial planning capabilities of multimodal large language models (MLLMs). VSP consists of various tasks, including maze navigation, block manipulation, collision detection, and pathfinding in real-world maps. These tasks are further broken down into subtasks focusing on perception and reasoning.

**Strengths:**

1. Necessart Benchmark:  spatial planning is one of the most important ability for the MLLM and VSP provides a framework for assessing MLLMs’ ability to understand and manipulate spatial environments;
2. This paper breaks down tasks into perception and reasoning subtasks, which helps identify specific bottlenecks that limit model performance in spatial planning.
3.  several leading MLLMs are evaluated, exposing room for the future improvements
4. Open-sourced dataset as mentioned in the paper

**Weaknesses:**

1. The current evaluation is not with dynamic and realistic environments and a more extensive evaluation in these settings would be beneficial. Some possible trials on auto driving or robotic manipulation is welcome.
2. Though this paper is mainly on LLM, some quick comparison with the traditional/classic planning methods is also welcome, at least could show some potential improvement room.
3. Limited analysis of model internals, further analysis on the performance gap or other factors will help the authors understand.

**Questions:**

1. How can the benchmark be adapted to evaluate other forms of spatial planning, such as 3D environments or object manipulation?

2. How can the findings from VSP inform the development of new MLLM architectures and training methods?

3. How can VSP be integrated into the development process of MLLMs to guide their design and optimization?

---

### Official Review · Reviewer_tYBX · 2024-11-03

**Soundness:** 3
**Presentation:** 3
**Contribution:** 4
**Rating:** 8
**Confidence:** 4

**Summary:**

This is a benchmark dataset paper. It particularly focuses on testing spatial planning capability of multimodal LLMs. The paper shows that existing multimodal LLMs fail to generate effective plans for even simple spatial planning tasks. The benchmark will encourage researchers to make their VLMs more spatial information aware.

**Strengths:**

- The paper focuses on spatial reasoning capabilities, also relevant to multiple real-world scenarios in robotics.
- Four different scenarios including 2D Maze Navigation, 3D Blocks World and Collision, as well as 2D Google Map were used for the evaluation.
- The supplementary material with detailed prompts and procedures is helpful.
- Experimental results confirming the weakness of the existing VLMs is meaningful.

**Weaknesses:**

- The selection of the VLMS tested now seems very slightly outdated. For the final version, maybe include newer models like Qwen2-VL, Claude-3.5, LLaVA-OneVision, BLIP-3, and so on?

**Questions:**

Any further discussions on how this benchmark could be extended further in the future?

---

### Official Review · Reviewer_oHsn · 2024-11-05

**Soundness:** 3
**Presentation:** 3
**Contribution:** 1
**Rating:** 3
**Confidence:** 3

**Summary:**

This paper build a new perception and spatial planning benchmark for multimodal LLMs. The key insights are using different visual planning tasks such as Maze Navigation and let LLMs solve these tasks by reasoning. They test the performance of some SOTA open-sourced and closed sourced MLLMs and find that GPT-4o is the best in their tasks for perception and spatial planning. However, MLLMs still have flaws in both perception and reasoning, the deficiency in the former capabilities is significantly worse.

**Strengths:**

1. This paper has a good dataset and benchmark contribution. The authors said they will open source the dataset.

2. This paper is well-written, especially its task defination sections.

**Weaknesses:**

1. As I know, the tasks you choose in your papers have already been proposed by many papers. I don't think you are the first in this area. See these references: [1],[2]

2. You state "The first two scenarios, Maze Navigation and Blocks World, are basic environments developed from classical planning tasks. Additionally, we challenge the model’s abilities in dynamic and realistic applications through the Collision and Google Map scenarios, respectively. All environments are fully observable through input images."

- But why are these tasks important for LLMs? Besides MLLMs, there are many other models that can be used for these tasks. I don't think these tasks  are necessary for MLLMs. Just like the navigation, we can easily transfer the question into language format. Thus LLMs can solve these questions in another way.

3. I am really confused with the concept of visual spatial planning in your paper. Most of the time we say visual spatial relations in the human's vision. If my understanding is right, it should be visual spatial relations in perception, and then make planning by reasoning. To better design your task, you should first let the model output the scene description for perception. And then guide the MLLMs to plan for your subtask. (Figure 2 in your paper.)

4. Maybe you should compare these MLLMs with humans in Table 2, thus we could know the importance of your task design. If humans are also hard to solve, I don't think these tasks  are necessary for MLLMs.

5. We may need more details for Table 2. For example, how many images you input for each models? Are you using different system prompts (not each input prompt) for these models? What is the scale of these open-sourced models? (7B, 34B, or larger)

REF:
[1] Ghosal, Deepanway, et al. "Are Language Models Puzzle Prodigies? Algorithmic Puzzles Unveil Serious Challenges in Multimodal Reasoning." arXiv preprint arXiv:2403.03864 (2024).
[2] Chia, Yew Ken, et al. "PuzzleVQA: Diagnosing Multimodal Reasoning Challenges of Language Models with Abstract Visual Patterns." arXiv preprint arXiv:2403.13315 (2024).

**Questions:**

I have raised most of the questions in the Weaknesses section.

**Details Of Ethics Concerns:**

No ethics concerns.

---

### Note · Authors · 2024-11-15

I have read and agree with the venue's withdrawal policy on behalf of myself and my co-authors.